# Scalloping of the Liver and Spleen on Preoperative CT-Scan of Pseudomyxoma Peritonei Patients: Impact on Prediction of Resectability, Grade, Morbidity and Survival

**DOI:** 10.3390/cancers14184434

**Published:** 2022-09-13

**Authors:** Vahan Kepenekian, Amaniel Kefleyesus, David Keskin, Nazim Benzerdjeb, Isabelle Bonnefoy, Laurent Villeneuve, Omar Alhadeedi, Abeer Al-Otaibi, Alexandre Galan, Olivier Glehen, Julien Péron, Pascal Rousset

**Affiliations:** 1Surgical Oncology Department, Hôpital Lyon Sud, Hospices Civils de Lyon, 69495 Pierre Bénite, France; 2CICLY-EA3738, Université Claude Bernard Lyon I (UCBL1), 69622 Lyon, France; 3Department of Visceral Surgery, Lausanne University Hospital CHUV, University of Lausanne (UNIL), 1015 Lausanne, Switzerland; 4Department of Radiology, Hôpital Lyon Sud, Hospices Civils de Lyon, 69495 Pierre Bénite, France; 5Department of Pathology, Hôpital Lyon Sud, Hospices Civils de Lyon, 69495 Pierre Bénite, France; 6Medical Oncology Department, Hôpital Lyon Sud, Hospices Civils de Lyon, 69495 Pierre Bénite, France; 7Laboratoire de Biométrie et Biologie Evolutive, Equipe Biostatistique-Santé, Université Claude Bernard Lyon I (UCBL1), 69622 Lyon, France

**Keywords:** pseudomyxoma peritonei (PMP), scalloping, resectability, prognostic factor, recurrence prediction, severe complications prediction

## Abstract

**Simple Summary:**

Liver and/or spleen scalloping is a common radiologic sign visible on preoperative computed tomography scans of pseudomyxoma peritonei patients. While several studies have reported a predictive value of this scalloping regarding resectability, histologic grade, postoperative morbidity risk and long-term oncologic outcomes, few data are available regarding splenic scalloping. The predictive value of hepatic and splenic scalloping characteristics (measures and density) was assessed. It appeared that scalloping was associated for a more extensive disease, requiring more complex cytoreduction, but it was not associated with resectability, histologic grade, postoperative complications, or survival.

**Abstract:**

Pseudomyxoma peritonei (PMP) is ideally treated by cytoreductive surgery (CRS) with hyperthermic intraperitoneal chemotherapy (HIPEC), leading to significant morbidity. Beyond the histologic grade, the prognosis lies in the completeness of cytoreduction (CC-score of 0/1 vs. 2/3) and the severe complication rate. The mucinous nature of the peritoneal implants sometimes induces liver and/or spleen scalloping on imaging. The predictive value of scalloping was assessed regarding resectability, grade, survival and severe morbidity. This monocentric, retrospective analysis compared CC-0/1 with CC-2/3 groups regarding liver and spleen scalloping parameters, assessed on pre-operative computed tomography (CT) scan, reviewed for the study. In addition, prognostic factors of severe complications and recurrence-free and overall survivals were explored in the CC-0/1 population. Overall, 129 patients were included (109 CC-0/1, 20 CC-2/3), with 58 (45%) exhibiting scalloping. All patients with splenic scalloping also had a liver one. Scalloping was more frequent (75% vs. 39%), with greater median maximal depth (21 vs. 11 mm) and higher PCI (32 vs. 14) in the CC-2/3 population, but was not predictive of either grade or survival. In CC-0/1 patients, survivals and postoperative complications were not affected by scalloping parameters. Scalloping appeared as a marker of advanced PMP, but was not predictive of grade, severe complications, or long-term outcomes.

## 1. Introduction

Pseudomyxoma peritonei (PMP) is a rare disease characterised by the dissemination of mucinous ascites and implants throughout the peritoneal cavity according to the redistribution phenomenon, originally issued from the rupture of a primary tumour, usually appendicular [1,2]. The comprehensive treatment combining a cytoreductive surgery (CRS) with hyperthermic intraperitoneal chemotherapy (HIPEC), developed initially by Sugarbaker, transformed the prognosis of such patients [3,4,5,6,7]. With systemic dissemination being very rare [8], that prognosis is driven by two main factors: the post-CRS residual disease, rated by the Completeness of Cytoreduction score (CC-score) and the pathologic characteristics defining low- and high-grade disease [4,9,10,11]. As patients often present with diffuse disease, extensive resections are required to reach a complete CRS, including multiple digestive and peri-hepatic resections, leading to high rates of severe postoperative complications, which are prone to affecting patients’ quality of life and prognosis [12,13,14,15,16,17,18,19,20]. Despite a complete CRS and HIPEC, half of high-grade patients and a quarter of low-grade ones recur [16,18,21]. Thus, predicting the level of resectability, the histologic grade, the risk of severe complications, the risk of recurrence, and the overall survival would help to tailor the treatment strategy and to better inform patients.

While peritoneal magnetic resonance imaging (MRI) has demonstrated advantages in the preoperative staging of mucinous tumours, contrast-enhanced computed tomography scan (CT-scan) remains the recommended imaging modality, and the most widely used, for preoperative evaluation [11,22,23]. Radiological features of PMP are unique, and several scores have been proposed for predicting postoperative outcomes based on mucinous tumour dimensions or on region of interest (ROI) density ratio measurements [24,25,26,27,28,29,30,31]. Among these signs, liver scalloping is frequently used, as it is easily detectable and measurable. It was defined by Seshul and Coulam in 1981 as an extrinsic pressure on the liver margin by adjacent peritoneal implants without liver parenchymal metastases [32]. Scalloping, included when used as part of a score and when used alone, has been reported to be predictive of resectability or survival, as in the Hotta et al. series, where liver scalloping was predictive of recurrence [24,29,30]. As the spleen can also exhibit such peritoneal involvement, we retrospectively explored the predictive value of these signs, in combination with different ratios of density of certain ROI, regarding the prediction of resectability, histologic grade, occurrence of severe postoperative complications, and long-term oncologic outcomes. The analysis allows us to conclude that the presence of scalloping pleads in favour of more advanced disease; however, this was not independently predictive of postoperative outcomes.

## 2. Materials and Methods

### 2.1. Study Framework

As part of RENAPE (the French network of peritoneal surface malignancies expert centres), our patients with rare peritoneal disease were included in a prospective database with a continuous follow-up [32]. Patients signed their informed consent, and the database was approved by the local ethics committee.

Consecutive PMP patients treated with CRS between January 2009 and December 2020 were included in this study according to the following selection criteria: pathologically confirmed PMP of appendicular origin, having never undergone systemic treatment, with a peritoneal carcinomatosis index (PCI) greater than 0 (exclusion of appendicular mucocele without peritoneal mucin), with consistent follow-up data, and with a preoperative contrast-enhanced CT-scan available for review. In cases where there were several CRS for the same patient, only the first was considered.

The treatment strategy was based on clinical examination looking for obstructive signs, preoperative work-up assessing operability and resectability, and was validated through a multidisciplinary team (MDT) meeting that included surgeons, oncologists, radiologists, and pathologists, all specialized in management of peritoneal surface malignancies. The CRS was performed as previously described [3,33]. Exploration of the peritoneal cavity allowed the quantification of peritoneal disease according to the PCI [8]. Then, a combination of peritonectomies and visceral resections was performed to achieve the minimal amount of residual disease. This result was graded according to the CC-score as follows: CC-0 when no residual lesions were visible, CC-1 if tumours of less than 2.5 mm remained, and CC-2 and 3 for residual disease of more than 2.5 mm and 2.5 cm, respectively [8]. In selected patients, CRS was followed by HIPEC, performed with the closed-abdomen technique with a combination of cisplatin and mitomycin C at 42 °C for 90 min.

Data regarding patient characteristics were collected along with the pathologic diagnosis established according to the PSOGI classification of low- or high-grade disease [9]. For patients graded according to the former classifications, a pathologic review was performed to re-stage the patients.

For the purpose of the analysis, treatment-related data were collected, notably the complications occurring within 90 postoperative days. The National Cancer Institute’s Common Terminology Criteria for Adverse Events (NCI-CTCAE) v5.0. was used to rate these complications, with grades 3, 4 and 5 being considered as severe [34].

Patients were followed-up postoperatively every 3 months for 2 years, then every 6 months for 3 years, then annually with clinical exam, serum tumour marker dosing (CEA, CA19-9 and CA125) and thoraco-abdomino-pelvic CT-scan (or peritoneal MRI and thoracic CT-scan if contra-indicated for iodine injection). Where suspected, recurrences were confirmed during the MDT meeting or with a radiologic or surgical biopsy in equivocal situations.

### 2.2. Preoperative CT-Scan Protocol and Image Analysis

The standard-dose CT acquisitions of the abdomen and pelvis were performed using multidetector CT units: Brilliance 40 (Phillips, Amsterdam, The Netherlands) before 2017 and Philips Ingenuity Elite 64 after 2017, Philips Healthcare; or Somatom Definition Flash (Siemens, Munich, Germany) 128, Siemens Healthcare after 2017; or Speed Light 16 General Electric Healthcare (Chicago, IL, USA) before 2017. Intravenous administration of contrast material was performed at the portal venous phase (Xenetix 300, Guerbet, Aulnay, France or Iomeron 400, Bracco, Italy). No oral contrast material or water preparation was used. The scanning parameters were 120 kVp with automatically set mAs values. Section images of 1 mm of the abdominopelvic region were obtained with a standard reconstruction algorithm with a thickness of 3 mm.

When performed elsewhere, multidetector CT examination protocols included at a minimum intravenous administration of contrast material at the portal venous phase and a slice thickness ≤ 5 mm.

A board-certified radiologist with 10 years of experience in peritoneal malignancies imaging and a radiologist with 3 years of experience in gastrointestinal oncology imaging, blinded to clinical information except for the knowledge of PMP, analysed the images by consensus on viewing workstations (Centricity Universal Viewer, General Electric Healthcare).

Presence of hepatic or splenic scalloping was first reported. When scalloping was equivocal on axial slice images, coronal and sagittal multiplanar reconstruction images were studied (Figure 1 and Figure 2).

The maximal thickness of mucin deposition corresponding to the depth of scalloping was evaluated using two methods on axial images: either the whole thickness of mucin deposition up to the diaphragm, as in Hotta et al. [24], or the thickness of mucin deposition up to the expected liver or spleen surface (i.e., intra-parenchyma part), as shown in Figure 1. When several lesions were available for interpretation, the deepest was selected for analysis. The maximal length of scalloping was also reported.

Finally, a quantitative analysis of tumour enhancement was performed. Regions of interest (ROIs) for measuring Hounsfield units (HU) were placed in the most attenuated component over the hepatic scalloping using a narrow contrast window. The ROIs were also placed in the liver parenchyma and the abdominal aorta to provide a reference value for the calculation of enhancement ratios. Intraparenchymal liver vessels were avoided when selecting the ROIs.

### 2.3. Cytoreductive Surgery and HIPEC

The CRS was performed as previously described [3,35]. After removing the mucinous ascites, the exploration of the peritoneal cavity allowed for the quantification of peritoneal disease according to the PCI [9]. A combination of peritonectomies and visceral resections was then performed to achieve the minimal amount of residual disease. This result was graded according to the CC-score as follows: CC-0 when no residual lesions were visible, CC-1 if tumours of less than 2.5 mm remained, and CC-2 and 3 for residual disease of more than 2.5 mm and 2.5 cm, respectively [9]. Among the peritonectomies, the removal of the entire hepatic capsule (glissonectomy) was performed in case of tumour deposit invasion with the technique of digital glissonectomy [36]. In selected patients, CRS was followed by HIPEC, performed via the closed-abdomen technique with a combination of cisplatin and mitomycin C at 42 °C for 90 min.

### 2.4. Statistical Analysis

A two-step analysis of the prediction value of scalloping was produced by computing several radiologic signs: presence or absence of liver scalloping, presence or absence of splenic scalloping, maximal depth of each scalloping, according to the two defined methods for the liver scalloping, the sum of maximal length and depth and two ROIs ratio of density (liver scalloping over liver parenchyma, liver scalloping over aorta).

Firstly, the ability to predict an incomplete CRS (CC-2/3) and the histologic grade was evaluated in the entire study population, with calculation of sensitivity, specificity, positive (PPV) and negative (NPV) predictive value of the presence of liver or splenic scalloping. Liver and splenic scalloping depth was computed to calculate the optimal cut-off based on median survival outcome and optimal Youden index. The R package “cutpointr” was used for this calculation. Secondly, the impact of scalloping on overall (OS) and recurrence-free survivals (RFS) and on severe postoperative complications was assessed in the population of patients completely resected (CC-0/1). The OS was defined as the time between the day of CRS and death and RFS as the time between the day of CRS and the diagnosis of recurrence or death, whichever occurred first. Survival rates were estimated using the Kaplan–Meier method and compared with the log-rank test. Overall survival and RFS medians were expressed by means of interquartile range (IQR).

The univariate hazard ratios (HR) for OS and RFS were estimated with 95% interval confidence (95% CI) through Cox regression analysis. Univariate odds ratio (OR) for severe postoperative morbidity were estimated with 95%CI through logistic regression analysis. Performance analyses (Se, Sp, PPV, PNV) were performed with the R package “epiR” (R package version 2.47).

Statistical analysis was performed using RStudio Software (RStudio: Integrated Development for R. PBC, Boston, MA, USA, 2020). Statistical significance was reached with two-sided *p*-value < 0.05.

## 3. Results

Between 2009 and 2020, 220 patients were treated in our referral centre for a PMP, of whom 129 met the inclusion criteria and were included in the study (Figure 3).

### 3.1. Pseudomyxoma Peritonei: Complete vs. Incomplete Cytoreduction

#### 3.1.1. Population Characteristics

In our population, 109 patients had a CC-0/1 resection and 20 a CC-2/3 CRS (Table 1). Patients in the CC-2/3 group were older and more often men, with a higher median PCI (32 (30–34) vs. 14 (6–24), respectively (*p* < 0.001)). The histology was of low grade in 91 (79%) of CC-0/1 patients, whereas it was of high grade in 13 (93%) CC-2/3 patients (*p* < 0.001). The median levels of the three serum tumour markers were significantly higher in the unresectable group. The surgery was logically longer in the completely resected patients and followed by HIPEC in 103 (94%) patients. No difference was observed in major postoperative complications rate and length of hospital stay.

#### 3.1.2. Predictive Value of Scalloping on Resectability and Histologic Grade

The median interval time between CT-scan and CRS was 41 days (21–80). Overall, 58 patients (45%) exhibited scalloping (Table 1). No patients exhibited splenic scalloping without liver scalloping, leading to 18 patients with scalloping on liver only and 40 patients with scalloping on both organs. Significantly fewer patients presented scalloping on preoperative CT-scan in the group of patients who had benefited from a CC-0/1 resection compared to CC-2/3 patients: 39% vs. 75%, respectively (*p* = 0.003).

The Se, Sp, NPV and PPV associated with the presence of scalloping regarding the prediction of resectability were 75%, 61%, 26% and 93%, respectively (Table 2). When considering the prediction of the histologic grade, the same parameters resulted in 58%, 57%, 38% and 75%, respectively.

When using the cut-off of scalloping maximal depth suggested by the ROC curves, the sensitivity of splenic scalloping deeper than 10mm for incomplete resection increased to 91%, and the specificity of hepatic scalloping deeper than 20 mm for a high grade disease increased to 93% (Table 2).

Among the 58 patients who presented scalloping on preoperative imaging, the median maximal depth of liver scalloping was 21 mm in CC-2/3 patients and 11 mm in CC-0/1 patients (*p* = 0.007). This difference was also statistically significant when considering the Hotta measurement technique (23 mm vs. 14 mm, *p* = 0.038) (Table 3). The splenic scalloping median maximal depth and length were not different between the two groups, but their ratio was higher in the CC-2/3 group: 0.4 (0.30–0.68) vs. 0.3 (0.20–0.35), respectively (*p* = 0.013). When combining the measurement of maximal dimensions of liver and splenic scalloping, the depth was significantly higher in the CC-2/3 group, while no differences were observed for the length and for the ratio maximal depth/length (Table 3).

No threshold of liver scalloping depth, nor liver + splenic depth appeared predictive of unresectability with sufficient discrimination power to be useful in daily clinical practice. The estimation of the Youden index was associated with wide variability (data not shown).

The three ratios of density of ROI did not lead to significant differences between completely and incompletely resected groups (Table 3).

#### 3.1.3. Scalloping Prognosis Impact on Overall Survival

The median follow-up time was 48 months (95% CI, 35.9–55.9). The median OS were not reached (NR) and 39.2 months (95% CI, 23.1-NR) in CC-0/1 and CC-2/3 patients, respectively, with a probability of survival at 5 years of 89% (95% CI 81–98%) and 24% (95% CI, 7–77%), respectively (*p* < 0.001). When comparing scalloping-positive and -negative patients, median OS was NR in both groups, but the probability of survival at 5 years was 71% (95% CI, 56–90%) and 87% (95% CI, 78–98%), respectively, *p* = 0.205 (Figure 4). A shift between the two curves was observed from the end of the second postoperative year, but was not significant after comparison with the log-rank test.

### 3.2. Scalloping Predictive Value in the Completely Resected Population (CC-0/1)

#### 3.2.1. Population Characteristics

The cohort of complete CRS was made up of 109 CC-0/1 patients, distributed into 66 patients without any scalloping and 43 patients with liver scalloping, of whom 29 also had splenic scalloping (Table 1). No differences in terms of age, gender, proportion of high grade and of HIPEC administration were found between patients with and without scalloping. However, similarly to the comparison of CC-0/1 to CC-2/3 patients, median PCI, serum tumour marker levels and operative time were higher in patients with scalloping, whose length-of-stay was also longer (Table 1). More patients in the group with scalloping required a splenectomy than in the group without (74% vs. 26%, respectively, *p* < 0.001). Overall, 52 out of 109 (48%) completely resected patients presented a severe postoperative complication, of whom 21 (49%) had scalloping.

#### 3.2.2. Prognostic Impact of Scalloping on Overall and Recurrence-Free Survivals and Severe Complications

The presence of scalloping in the CC-0/1 population was not predictive of OS, with a probability of survival at 5 years of 85% (95% CI, 71–100%) in the case of scalloping and 91% (95% CI, 82–100%) without scalloping, *p* = 0.82 (Figure 5).

Following the first CRS with complete resection, 27 (26%) patients exhibited recurrence. The 5-year RFS was 72% (95% CI 63–83%). Patients with and without scalloping did not reach the median RFS, with 5-year RFS of 74% and 72%, respectively (*p* = 0.66) (Figure 5).

In univariate analysis, a CA-125 level over 41 UI and a PCI > 25 were associated with a worse OS; high CA-125 and high histologic grade were associated with a worse RFS; and no factor was associated with the risk of severe postoperative complications. None of the scalloping measurements were significantly associated with either RFS or OS (Table 4). Multivariate analysis did not reveal any independent prognostic factors. Based on the high variance of the estimated Youden index values and the graphical examination of the ROC curves, it was established that the overall discrimination value of radiological measurements for predicting disease recurrence was low, and no optimal cut-off value could be established (data not shown).

## 4. Discussion

In this analysis, liver and spleen scalloping observed on preoperative CT-scan of PMP patients appeared to be a marker of more aggressive disease, being twice as frequent in unresectable patients; however, it was not predictive of resectability or of overall or recurrence-free survivals. Similarly, no threshold in scalloping measurements and density of ROIs was found to be useful in clinical practice for consistently predicting these outcomes.

In the completely resected population, the presence of any type of scalloping was associated with higher PCI, higher levels of tumour markers, and longer CRS and length-of-stay, suggesting that it is associated with more complex surgeries. Consistent with the principle of mucin redistribution within the peritoneal cavity, the frequency of hepatic scalloping increased with PCI, and all patients with splenic scalloping also had hepatic scalloping. The presence of the two locations of scalloping is thus a marker of a more extensive disease. Moreover, the presence of a splenic scalloping justifies a splenectomy, which is more frequently performed in this group. These two elements lead, in principle, to a poorer prognosis and a higher risk of severe complications, even if our results did not confirm that, probably as a result of lack of statistical power. In any case, the presence of scalloping, particularly splenic, is an additional argument in favour of referring the patient to an expert centre.

PMP is a unique peritoneal disease with dedicated management rules inspired by Sugarbaker’s model and refined over the previous decades in expert centres [3,6,7,11,17,37]. Mucin is the cornerstone of PMP pathogenesis, inducing a constellation of specific radiologic signs like the redistribution phenomenon, the smudged omentum, ‘cauliflowering’ of the small bowel, mucinous implants, and scalloping, commonly of the liver or the spleen [1,29]. The description of these signs is well established, and several authors have aimed to assess the predictive value of the key elements of PMP management: resectability, histologic grade, and the risk of severe postoperative complications [24,25,26,27,29,30,31,38].

For such predictions to be useful, they should allow a shift in treatment strategy that is likely to improve the overall prognosis. In this regard, predicting the histologic grade preoperatively is of limited interest so far, knowing that no strategy of neo-adjuvant systemic chemotherapy is recommended for high-grade PMP patients, as long as they are upfront resectable [4,11,39,40,41]. It could, however, be useful for frail patients to discuss the opportunity of performing an extensive surgery, in consideration of a poorer prognosis. On the other hand, resectability is a major prognostic factor for PMP patients, and anticipating the level of resectability could influence the treatment strategy [4,11,29,30,31]. Achieving a complete CRS in high-peritoneal-load disease requires extensive cytoreduction, increasing the risk of severe postoperative complications and the deterioration of quality of life [12,13,14,15,19,20]. Predicting the risk of unresectability allows for appropriate surgical planning, based on maximal tumour debulking or, for some authors, on a two-step CRS [42,43]. Several works have evaluated strategies for predicting resectability. In particular, Bouquot et al. proposed a score based on the mucin thickness measurement of five peri-hepatic areas, also performed on preoperative CT-scan [26]. Established with a rigorous methodology, the score appeared to be higher in non-resectable patients. When using a sum threshold of 28 mm, the sensibility and the specificity were 80 and 69%, respectively. Concretely, this score is based on the peri-hepatic invasion by mucinous implants, which is clearly a specific feature of PMP and the main limit of resectability in these patients. However, the 20% false negatives nevertheless justifies surgical exploration, considering the prognostic impact of missing an initial window of resectability in PMP.

Subsequently, Hotta et al. proposed the use of another radiological sign specific to PMP to predict a CC-0/1 CRS: liver scalloping depth [24]. This measurement was performed in 64 patients treated with CC-0/1 resection, 62.5% of whom presented liver scalloping. The presence of this sign was independently associated with a poorer RFS in multivariate Cox analysis (HR 3.1 (IC 95% 1.1–8.8), *p* = 0.031). A threshold of 20 mm discriminated patients with a significantly higher risk of recurrence. Our approach was similar to that of Hotta et al.; however, our results were slightly different, which can be explained by there being some differences in both patient characteristics and in methodology. The Hotta et al. series was composed of more advanced disease, with 62.5% of patients exhibiting liver scalloping, 52% possessed high-grade disease, and a median PCI of 28.5, while we reported 37% of patients with scalloping, 25% with high-grade disease, and a median PCI of 13 (6–24) [24]. In addition, the technique of scalloping measurement was different in the Hotta et al. series, resulting in a population with larger scalloping. Finally, Hotta et al. focused on a shorter study period and were able to use more recent and therefore more performative CT machines, allowing for more accurate reconstructions and measurements. Notably, between 2009 and 2017, in our study, some patients underwent examination on a 16-detector-row CT, which has a more limited resolution than that used for current techniques. Although this technology enabled correct multiplanar reconstructions, it may have induced a possible bias of measurement. Thus, more advanced disease and larger scalloping led to a significant association with poorer RFS. Consequently, both series finally concurred that the presence of scalloping was a marker of more advanced disease with, overall, a poorer prognosis.

Nonetheless, while the CC-score (0/1 vs. 2/3) was confirmed as a determinant prognostic factor of OS, the presence of scalloping was not [4]. This result echoes a former observation made by Sun et al. [25]. Despite the absence of identification of completely and incompletely resected patients in their series, their analysis revealed that, in the 55 patients with hepatic scalloping, the maximum thickness was lower in high-grade patients when compared to low-grade ones (11 vs. 20 mm, respectively, *p* = 0.021). The comparison with our results is rendered difficult by the difference in pathology (i.e., the high proportion of non-appendiceal PMP in the Sun et al. series) and by the unknown CC-score. However, these results seem to indicate a propensity of low-grade disease to induce larger scalloping, an observation already described previously [44]. This phenomenon could partly explain why scalloping was not predictive of survival: the detrimental effect on the survival with high PCI was compensated by the fact that larger scalloping was more associated with low-grade disease, which is, conversely, a good prognostic factor.

These observations could be put into perspective by the mucin consistency angle of analysis of PMP behaviour. Morris et al. defined three types of mucin with respect to hardness index (soft, semi-hard and hard), which correlated with survival [45,46]. It is possible that variations in mucin consistency are also responsible for the heterogeneity of of the impact of scalloping on the assessed outcomes. Hard mucin could lead to thinner scalloping, while it is also associated with a more aggressive biology, and thus to a poorer prognosis. However, the contrasting results of the different series assessing scalloping thickness and density impact on the main outcomes could finally mean that this sign is not sufficient in itself to draw conclusions, but must be integrated into a multimodal analysis [28,29].

Our work presents several limitations, mainly linked to its retrospective nature. The preoperative and intraoperative resectability evaluation is multifactorial and evolves with time, although it influences long-term oncologic outcomes. Moreover, the heterogeneity of studies reporting outcomes of CRS-HIPEC for this rare and complex disease, regarding pathology and surgical strategies, render direct comparisons difficult.

## 5. Conclusions

In conclusion, liver and splenic scalloping are radiologic signs frequently encountered in PMP patients’ preoperative work-up, and correlated with more advanced disease. However, the inconstant association of scalloping with long-term outcomes from one series to another suggest that this sign is not solely reflective of PMP behaviour. Scalloping could therefore be considered as a secondary sign of advanced disease, but not sufficient on its own to tailor treatment strategy while awaiting the development of radiomics [47], which will hopefully overcome these limitations.

## Figures and Tables

**Figure 1 cancers-14-04434-f001:**
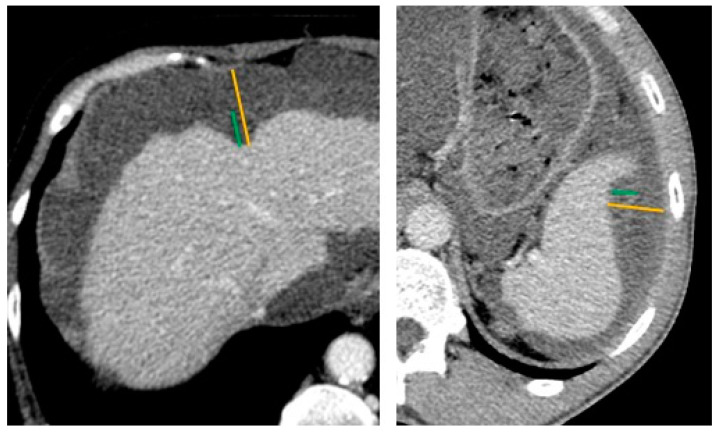
Technique of scalloping measurement on computed tomography scan (CT-scan). Axial contrast-enhanced with a narrow-window CT image showing liver (**left**) and spleen (**right**) scalloping in two patients with pseudomyxoma peritonei. The yellow lines indicate the maximum thickness of mucin deposition up to the diaphragm, and the green lines indicate the maximum thickness of mucin at the expected liver or spleen surface.

**Figure 2 cancers-14-04434-f002:**
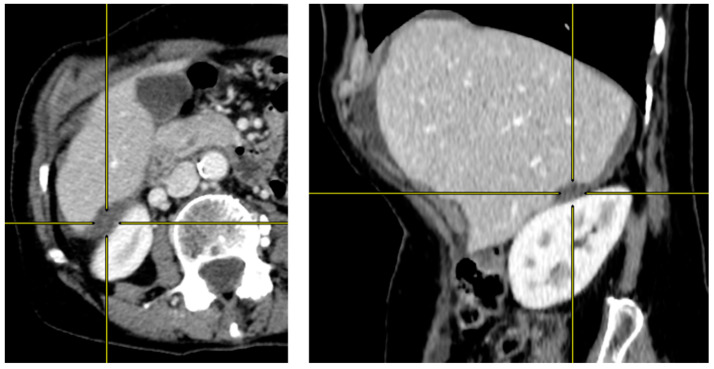
Multiplanar reconstruction for assessing equivocal liver scalloping on axial image. Axial contrast-enhanced CT image (**left**) showing an equivocal liver scalloping of intraperitoneal mucin deposition in the Morison pouch. Sagittal reconstruction CT image (**right**) showing a clearer indentation of the liver, allowing a confident assessment of liver scalloping.

**Figure 3 cancers-14-04434-f003:**
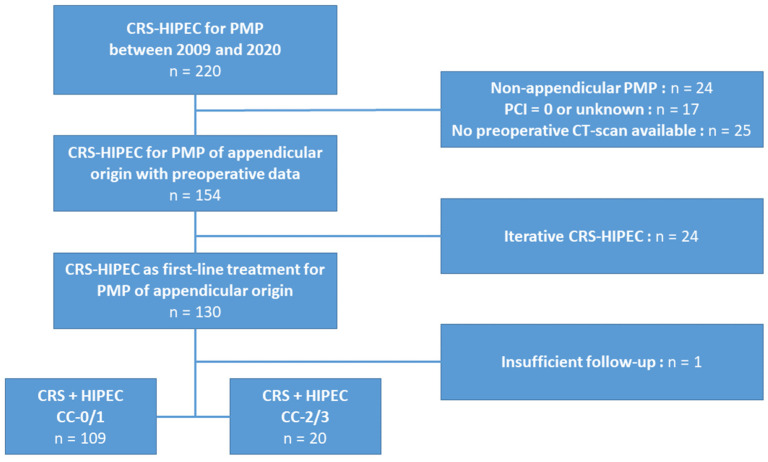
Study flowchart.

**Figure 4 cancers-14-04434-f004:**
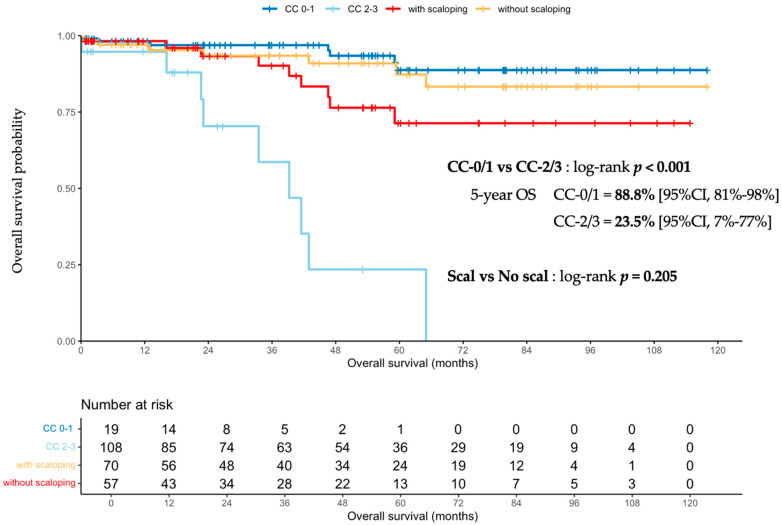
Overall survival probability in the overall population comparing completely to incompletely resected patients, and patients with scalloping to those without. CC-0/1, completeness of cytoreduction score of 0 or 1 corresponding to a complete cytoreductive surgery; OS, overall survival; Scal, scalloping.

**Figure 5 cancers-14-04434-f005:**
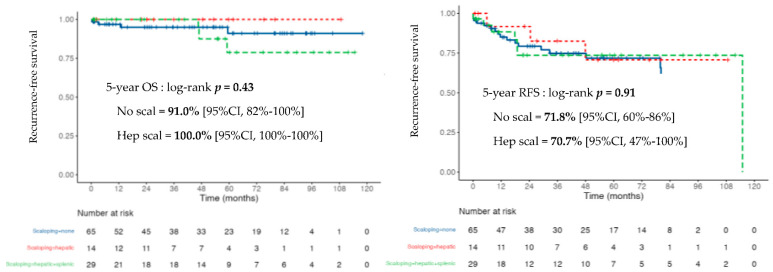
Overall and recurrence-free survival probability in the completely resected population (CC-0/1) by scalloping features.

**Table 1 cancers-14-04434-t001:** Patient characteristics.

	Overall Population				Completely Resected Population			
	CC-0/1 *N* = 109	CC-2/3 *N* = 20	Missing *N* (%)	*p* ^2^	With Scalloping *N* = 43	Without Scalloping *N* = 66	Missing *N* (%)	*p* ^2^
Age ^1^	57.1 [46.2, 65.4]	67.1 [58.0, 77.5]	0	0.003	55.6 [47.4, 65.5]	57.3 [40.7, 64.8]	0	0.7
Gender, male	38 (35%)	12 (60%)	0	0.034	12 (28%)	26 (39%)	0	0.2
ASA			19 (15%)	0.2			13 (12%)	0.4
1	44 (46%)	5 (36%)			21 (50%)	23 (43%)		
2	46 (48%)	6 (43%)			20 (48%)	26 (48%)		
3	6 (6.2%)	3 (21%)			1 (2.4%)	5 (9.3%)		
PCI ^1^	14.0 [6.0, 24.0]	31.5 [29.8, 33.8]	0	<0.001	24.0 [17.0, 27.0]	8.0 [3.2, 14.8]	0	<0.001
Histologic grade, high	18 (21%)	13 (93%)	30 (23%)	<0.001	9 (24%)	9 (19%)	24 (22%)	0.5
BMI (kg/m^2^) ^1^	23.8 [21.5, 27.9]	24.2 [22.5, 26.5]	8 (6.2%)	>0.9	22.7 [21.5, 26.2]	24.0 [21.8, 28.4]	6 (5.5%)	0.3
Splenectomy rate	50 (46%)	5 (26%)	1 (0.8%)	0.11	32 (74%)	18 (27%)	0	<0.001
CA-19.9 ^1^	15.0 [7.0, 36.0]	176.0 [77.0, 1,597.0]	32 (25%)	<0.001	28.0 [10.5, 52.8]	10.0 [6.0, 19.0]	28 (26%)	0.001
CEA ^1^	2.9 [1.3, 10.7]	53.4 [32.0, 147.0]	29 (22%)	<0.001	5.9 [2.2, 18.8]	1.8 [0.9, 3.6]	25 (23%)	<0.001
CA-125 ^1^	28.0 [17.2, 65.0]	85.0 [70.0, 168.0]	51 (40%)	<0.001	48.4 [27.2, 83.0]	20.0 [10.0, 28.7]	42 (39%)	<0.001
Operative time (min) ^1^	300.0 [240.0, 390.0]	210.0 [195.0, 225.0]	30 (23%)	0.026	330.0 [300.0, 420.0]	270.0 [210.0, 360.0]	17 (16%)	0.001
HIPEC	103 (94%)	5 (25%)	0	<0.001	41 (95%)	62 (94%)	0	>0.9
Severe complications (CD ≥ 3)	52 (48%)	10 (50%)	1 (0.8%)	0.9	21 (49%)	31 (48%)	1 (0.9%)	>0.9
Postoperative mortality	1 (0.9%)	1 (5.0%)	1 (0.8%)	0.3	0 (0%)	1 (1.5%)	1 (0.9%)	>0.9
Hospital length of stay ^1^	19.0 [14.0, 29.0]	17.5 [13.2, 21.8]	8 (6.2%)	0.5	23.0 [17.0, 33.0]	16.0 [12.0, 22.8]	6 (5.5%)	<0.001
Scalloping rate	43 (39%)	15 (75%)	0	0.003	NA			

ASA, American Society of Anesthesiologists; PCI, peritoneal carcinomatosis index; BMI, body mass index; CD, Clavien-Dindo score; NA, not applicable ^1^ Median [IQR]. ^2^ Wilcoxon rank-sum test; Pearson’s Chi-squared test; Fisher’s exact test.

**Table 2 cancers-14-04434-t002:** Contingency table assessing scalloping value for prediction of resectability and histologic grade.

	Scal (Any Kind) Y vs. N	Scal H + S vs. No Scal	Scal H > 20 mm	Scal S > 10 mm
**Resectability** **(CC-0/1 vs. CC-2/3)**				
Se	75%	55%	53%	91%
Sp	61%	73%	88%	28%
PPV	26%	28%	62%	32%
NPV	93%	90%	84%	89%
**Histologic grade** **(low vs. high)**				
Se	58%	42%	39%	8%
Sp	57%	72%	93%	89%
PPV	38%	41%	78%	33%
NPV	75%	73%	71%	59%

Scal, scalloping; Y, yes; N, no; H, hepatic; S, splenic; vs., versus; Se, sensitivity; Sp, specificity; PPV, positive predictive value; NPV, negative predictive value.

**Table 3 cancers-14-04434-t003:** Scalloping parameters by resectability and histologic grade.

Characteristics	Cytoreduction Completeness				Histologic Grade			
	CC-0/1 *N* = 109	CC-2/3 *N* = 20	Missing *N* (%)	*p* ^2^	Low *N* = 68	High *N* = 31	Missing *N* (%)	*p* ^3^
**PCI** ^1^	14.0 [6.0, 24.0]	31.5 [29.8, 33.8]	0	<0.001	14.0 [6.0, 24.0]	24.0 [18.0, 31.0]	0	<0.001
**Scalloping**			0	0.008			0	0.3
None	66 (61%)	5 (25%)			39 (57%)	13 (42%)		
Hepatic scal	14 (13%)	4 (20%)			10 (15%)	5 (16%)		
Hepatic + Splenic scal	29 (27%)	11 (55%)			19 (28%)	13 (42%)		
**Liver scalloping** ^1^								
max depth (mm)	11.0 [7.0, 17.0]	21.0 [11.5, 24.0]	71 (55%)	0.007	11.0 [7.0, 17.0]	13.5 [10.0, 24.5]	52 (53%)	0.2
max depth Hotta (mm)	14.0 [8.5, 27.0]	23.0 [18.0, 32.5]	71 (55%)	0.038	14.0 [7.0, 26.0]	15.5 [10.8, 25.0]	52 (53%)	0.4
max length (mm)	50.0 [32.5, 66.0]	60.0 [40.0, 80.0]	71 (55%)	0.3	48.0 [35.0, 70.0]	60.0 [43.2, 66.5]	52 (53%)	0.3
ratio max depth/length	0.2 [0.2, 0.4]	0.4 [0.2, 0.5]	71 (55%)	0.10	0.2 [0.2, 0.4]	0.3 [0.2, 0.4]	52 (53%)	>0.9
**Splenic scalloping** ^1^								
max depth (mm)	10.0 [8.0, 12.0]	12.0 [10.0, 15.0]	89 (69%)	0.3	12.0 [10.0, 13.5]	9.0 [7.0, 10.0]	67 (68%)	0.014
max depth Hotta (mm)	11.0 [9.0, 22.0]	17.0 [11.0, 25.0]	89 (69%)	0.3	14.0 [10.0, 23.5]	12.0 [7.0, 18.0]	67 (68%)	0.3
max length (mm)	40.0 [30.0, 47.0]	30.0 [20.5, 41.5]	89 (69%)	0.2	40.0 [30.0, 60.0]	28.0 [17.0, 35.0]	67 (68%)	0.019
ratio max depth/length	0.3 [0.2, 0.3]	0.4 [0.3, 0.7]	89 (69%)	0.013	0.3 [0.2, 0.4]	0.3 [0.3, 0.6]	67 (68%)	0.4
**Liver + Splenic scalloping** ^1^								
max depth (mm)	19.0 [16.0, 34.0]	32.0 [24.5, 38.0]	89 (69%)	0.040	27.0 [17.5, 35.0]	22.0 [16.0, 32.0]	67 (68%)	0.6
max length (mm)	91.0 [70.0, 110.0]	113.0 [70.0, 128.0]	89 (69%)	0.5	91.0 [70.0, 107.5]	88.0 [75.0, 113.0]	67 (68%)	0.9
ratio max depth/length	0.2 [0.2, 0.3]	0.3 [0.2, 0.4]	89 (69%)	0.065	0.3 [0.2, 0.3]	0.2 [0.2, 0.4]	67 (68%)	0.4
**ROI density (HU)**								
ratio ROI scal/aorta ≥ 0.3	8 (19%)	4 (27%)	71 (55%)	0.5	5 (17%)	4 (22%)	52 (53%)	0.7
ratio ROI scal/liver ≥ 0.6	13 (30%)	8 (53%)	71 (55%)	0.13	9 (31%)	8 (44%)	52 (53%)	0.4

PCI, peritoneal carcinomatosis index; ROI, region of interest; HU, Hounsfield unit. ^1^ Median [IQR]. ^2^ Wilcoxon rank-sum test; Fisher’s exact test. ^3^ Wilcoxon rank-sum test; Fisher’s exact test; Pearson’s Chi-squared test.

**Table 4 cancers-14-04434-t004:** Univariate analysis of predicting factors of overall and recurrence-free survivals and severe postoperative complications in completely resected patients.

		Overall Survival			Recurrence-Free Survival			Severe Complications		
	N	HR	95% CI	*p*	HR	95% CI	*p*	OR	95% CI	*p*
Age, ≥60 yo	108	0.55	0.11, 2.83	0.5	0.99	0.46, 2.16	>0.9	1.43	0.67, 3.09	0.4
Sex, Male	108	2.84	0.63, 12.8	0.2	0.91	0.40, 2.10	0.8	0.95	0.43, 2.10	>0.9
CEA, >15 IU	83	4.28	0.94, 19.4	0.059	2.50	0.94, 6.61	0.065	1.14	0.41, 3.30	0.8
CA-19.9, >20 IU	80	2.04	0.41, 10.1	0.4	1.67	0.62, 4.49	0.3	0.72	0.29, 1.80	0.5
CA-125, >41 IU	66	**6.47**	**1.23**, **34.1**	**0.028**	**3.17**	**1.08**, **9.32**	**0.036**	1.32	0.49, 3.59	0.6
Histologic grade, high	84	2.31	0.32, 16.5	0.4	**4.20**	**1.74**, **10.2**	**0.001**	2.06	0.72, 6.22	0.2
PCI, >25	108	**7.00**	**1.51**, **32.5**	**0.013**	1.37	0.47, 4.01	0.6	1.43	0.52, 4.06	0.5
**Scalloping**	108									
none		-	-		-	-		-	-	
hepatic		0.00	0.00, Inf	>0.9	0.78	0.23, 2.68	0.7	0.30	0.06, 1.06	0.083
hepatic + splenic		1.78	0.40, 7.94	0.5	0.88	0.35, 2.23	0.8	1.79	0.74, 4.49	0.2
**Liver scalloping**										
max depth, >15 mm	43	1.19	0.11, 13.1	0.9	1.12	0.28, 4.49	0.9	1.64	0.46, 6.17	0.5
max depth Hotta, >20 mm	43	1.33	0.12, 14.9	0.8	1.24	0.31, 4.99	0.8	1.64	0.46, 6.17	0.5
max length, >60 mm	43	4.28	0.39, 47.3	0.2	1.32	0.33, 5.28	0.7	1.33	0.36, 5.07	0.7
ratio max depth/length, >0.4	43	0.00	0.00, Inf	>0.9	0.48	0.06, 3.88	0.5	2.53	0.57, 13.6	0.2
**Splenic scalloping**										
max depth, >8 mm	29	0.21	0.02, 2.37	0.2	0.43	0.09, 2.13	0.3	0.16	0.01, 1.11	0.11
max depth Hotta, >20 mm	29	0.00	0.00, Inf	>0.9	0.56	0.06, 4.79	0.6	0.34	0.06, 1.72	0.2
max length, >40 mm	29	3.62	0.33, 40.3	0.3	0.96	0.17, 5.25	>0.9	0.32	0.06, 1.51	0.2
ratio max depth/length, >0.4	29	0.00	0.00, Inf	>0.9	0.00	0.00, Inf	>0.9	1.29	0.20, 10.7	0.8
**ROI density (HU)**										
ratio ROI scal/aorta ≥ 0.3	43	1.37	0.12, 15.2	0.8	1.52	0.38, 6.11	0.6	4.00	0.79, 30.0	0.12
ratio ROI scal/healthy liver ≥ 0.6	43	2.95	0.26, 32.9	0.4	1.37	0.36, 5.12	0.6	3.37	0.88, 14.9	0.085

yo, years old; HR, hazard ratio; CI, confidence interval; OR, odds ratio; PCI, peritoneal carcinomatosis index; scal, scalloping; ROI, region of interest; Inf, inferior to 0.

## Data Availability

Data are available on demand by email to the corresponding author.

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
