# Peer review of "Scalloping of the Liver and Spleen on Preoperative CT-Scan of Pseudomyxoma Peritonei Patients: Impact on Prediction of Resectability, Grade, Morbidity and Survival"

_cancers, 2022, doi:10.3390/cancers14184434_

Round 1

Reviewer 1 Report

The article has presented how the preoperative findings of scalloping of the liver and spleen by CT scan affect the resectability, grade, and morbidity of pseudomyxoma peritonei (PMP) patients in the French cohort. They found that scalloping was a marker of advanced PMP but was not predictive of grade or morbidity. It is an exciting article; however, there are some concerns about this article. 1. In the tile, what is the "oncologic outcome"? The authors should select more specific words or omit the sentences. 2. The authors could describe in more detail CRS and HIEC, including their own figures for the general readers. 3. The authors should state this study's primary and secondary outcomes in the introduction. 4. References are relatively old. 5. Moderate English edition is necessary.

Author Response

We thank the reviewer for his careful reading and relevant comments prone to improve our manuscript.

The article has presented how the preoperative findings of scalloping of the liver and spleen by CT scan affect the resectability, grade, and morbidity of pseudomyxoma peritonei (PMP) patients in the French cohort. They found that scalloping was a marker of advanced PMP but was not predictive of grade or morbidity. It is an exciting article; however, there are some concerns about this article.

  1. In the tile, what is the "oncologic outcome"? The authors should select more specific words or omit the sentences

Indeed the terminology “oncologic outcomes” is imprecise. We turned it into survival.

  1. The authors could describe in more detail CRS and HIEC, including their own figures for the general readers

An extended paragraph describing the technic of CRS and HIPEC has been added. To address one of the comment of the second reviewer, we had to add a figure related to the measurement technic of scalloping in difficult situations. The number of Figures being limited by the journal requirements, we preferred to maintain that list of figures, more directly related to the subject of this manuscript.

  1. The authors should state this study's primary and secondary outcomes in the introduction

A sentence has been added at the end of the introduction.

  1. References are relatively old

We selected the seminal references regarding PMP management and those related to imaging of PMP. As it is a rare condition, the relevant papers are not so frequent and could appear a bit old. However, some recent references has been added.

  1. Moderate English edition is necessary

The English has been reviewed and edited by a native speaker (OA).

Reviewer 2 Report

The authors present their experience in CT scan of pseudomyxoma peritonei with scalloping of the liver and spleen and its impact on prediction of respectability grade of the disease, morbidity and outcomes. The argument was deeply treated in pseudomixoma peritonei by Hotta in Europ Radiol 2020 doi: 10.1007/s00330-020-06756-2 as cited in the present paper. However the comparison with the experience of Hotta have to be stressed about eventual difference in results obtained. For instance the CT technique acquisition of the experience of Hotta is more accurately described and the technology up to date (16 row Ct is a little bit old). Example of MPR reconstruction images in complex situation would be helpful for readers.

Author Response

We thank the reviewer for his careful reading and relevant comments prone to improve our manuscript.

The authors present their experience in CT scan of pseudomyxoma peritonei with scalloping of the liver and spleen and its impact on prediction of respectability grade of the disease, morbidity and outcomes. The argument was deeply treated in pseudomixoma peritonei by Hotta in Europ Radiol 2020 doi: 10.1007/s00330-020-06756-2 as cited in the present paper.

However the comparison with the experience of Hotta have to be stressed about eventual difference in results obtained. For instance the CT technique acquisition of the experience of Hotta is more accurately described and the technology up to date (16 row Ct is a little bit old).

The following sentence has been added in the discussion where we discussed the difference between our results and Hotta’s study results: “Finally, Hotta et al focused on a shorter study period and was able to use more recent and therefore resolving CT units, allowing for more accurate reconstructions and measurements. »

Example of MPR reconstruction images in complex situation would be helpful for readers.

Such an example has been added in the figure 2, indeed usefully completing the figure 1 showing the technic of scalloping measurement.

Reviewer 3 Report

none

Author Response

No specific response needed there.

Thank you

Round 2

Reviewer 2 Report

The author addressed very briefly some of the points I have mentioned in my comments but acquisition technique in CT is stil poorly described with a wide difference od technology used and this may represent a bias in author experience; 16row CT is an old technology since at least 10 years they should mention as a possible bias measurement with this technology.

Author Response

We thank the reviewer for his comment and regret that she/he was not satisfied with our revision.

We tried to detail this in the dedicated part of the discussion highlighting the fact that some of our CT-scans were performed on a 16-row machine, could have interfered with the precision of our measurements.

However, considering that the differences would have concerned only a few millimeters, not including the entire cohort, it is unlikely that this bias would have significantly changed our overall results.

We hope that the reviewer will accept this new and extended revision.

Yours sincerely,
